# A broad-spectrum gas sensor based on correlated two-dimensional electron gas

Yuhao Hong[1], Long Wei [ID][1], Qinghua Zhang [ID][2], Zhixiong Deng[1], Xiaxia Liao[3], Yangbo Zhou[3], Lei Wang[1], Tongrui Li[1], Junhua Liu[1], Wen Xiao[1], Shilin Hu[1], Lingfei Wang [ID][4], Lin Li[1], Mark Huijben [ID][5], Yulin Gan [ID][1], Kai Chen [ID][1], Gertjan Koster [ID][5], Guus Rijnders [ID][5] ✉ & Zhaoliang Liao [ID][1,5] ✉

Designing a broad-spectrum gas sensor capable of identifying gas components in complex environments, such as mixed atmospheres or extreme temperatures, is a significant concern for various technologies, including energy, geological science, and planetary exploration. The main challenge lies in finding materials that exhibit high chemical stability and wide working temperature range. Materials that amplify signals through non-chemical methods could open up new sensing avenues. Here, we present the discovery of a broad-spectrum gas sensor utilizing correlated two-dimensional electron gas at a delta-doped $LaAlO_3$/$SrTiO_3$ interface with $LaFeO_3$. Our study reveals that a back-gating on this two-dimensional electron gas can induce a non-volatile metal to insulator transition, which consequently can activate the two-dimensional electron gas to sensitively and quantitatively probe very broad gas species, no matter whether they are polar, non-polar, or inert gases. Different gas species cause resistance change at their sublimation or boiling temperature and a well-defined phase transition angle can quantitatively determine their partial pressures. Such unique correlated two-dimensional electron gas sensor is not affected by gas mixtures and maintains a wide operating temperature range. Furthermore, its readout is a simple measurement of electric resistance change, thus providing a very low-cost and high-efficient broad-spectrum sensing technique.

A broad-spectrum gas sensor is highly in demand for many advanced applications, especially in chemical synthesis, geological science, and planetary exploration[1–3]. However, severe challenges arise from the complex gas composition in many actual scenarios where inert gas, polar gas, non-polar gas, organic gas, and non-organic gas coexist. For example, the capability to detect $CH_4$, $O_2$, $H_2O$, and other types of gas species in one device is required for exploring evidence of carbon-based life. On the one hand, sophisticated mass spectrometry systems can probe gas components, but they are very expensive, heavy, and occupy a large volume. For example, the Quadrupole Mass Spectrometer and the Tunable Diode Laser Absorption Spectroscopy equipped by Curiosity, which are the main tools for analyzing the composition of the Martian atmosphere, need to be equipped with a vacuum system and a purification system to remove $CO_2$ when they are working[2]. On the other hand, numerous metal-oxide-semiconductor (MOS) gas sensors exhibit cross-responses and limited selectivity of gases, and also typically only work for a few specific gases after doping, light or temperature modulation[4,5]. Although great efforts have been

[1]National Synchrotron Radiation Laboratory, University of Science and Technology of China, Hefei, China. [2]Beijing National Laboratory for Condensed Matter Physics, Institute of Physics, Chinese Academy of Sciences, Beijing, China. [3]School of Physics and Materials Science, Nanchang University, Nanchang, China. [4]National Research Center for Physical Sciences at Microscale, University of Science and Technology of China, Hefei, China. [5]MESA+ Institute for Nanotechnology, University of Twente, Enschede, the Netherlands. ✉e-mail: a.j.h.m.rijnders@utwente.nl; zliao@ustc.edu.cn

made to enhance the linear response of sensors in complex gas environments by combining MOS-based sensors into arrays[6] or using temperature modulation[7] or dielectric excitation[8], the use of mainstream MOS gas sensors is still limited due to huge temperature differences and harsh environment typical of outer space[4]. Furthermore, owing to the inherent stable chemical characteristics of inert and rare gases[9], their selection through conventional methods is very challenging. This situation further restricts the development of truly broadspectrum gas sensors.

The discovery of a high-mobility two-dimensional electron gas (2DEG)[10] at the interface between two insulators, $SrTiO_3$ and $LaAlO_3$, has extensively expanded the realm of correlated oxide electronics and brought groundbreaking technological application. The electronic state of the correlated 2DEG (C-2DEG) can be easily controlled by an electric field, indicating the potential for advancing oxide electronic devices. By taking the advantage of interfacial correlated 2DEG, several device concepts have been developed, especially single electron devices governed by a 'water cycle' mechanism[11], such as tunnel barriers[12,13], rectifying junctions[14,15], 'SketchFET' transistors[16] and photoconductive switches[17]. These devices were created and erased using top gate voltages applied by a conducting atomic force microscope (c-AFM) probe on approximately 3-unit cell (u.c.) thick $LaAlO_3$ films grown on $TiO_2$-terminated $SrTiO_3$ substrate[18]. Nevertheless, 2-DEG devices driven by back-gating are rarely reported, which would produce a completely different phenomenon than top gating[19].

Here, we present the exploration of a non-volatile mechanism of back-gate driven non-volatile depletion of interface 2-DEG gas in delta doped $LaAlO_3/SrTiO_3$ interface with $LaFeO_3$. With such unique carrier depleted interface, no matter inert gases (e.g. $N_2$), rare gases (e.g., Ar), polar gases (e.g. $CH_4$) or non-polar gases can be detected in one back-gated device. This unique broad-spectrum C-2DEG gas sensor based on a delta doped[20] LAO/STO interface is schematically shown in Fig. 1a. A 5-u.c. thick $LaFeO_3$ layer is inserted between $TiO_2$-terminated $SrTiO_3$ and a 10-u.c. thick $LaAlO_3$. All the films were epitaxially grown by our home-made PLD system. The as grown $LaAlO_3/LaFeO_3/SrTiO_3$ (LAO/ LFO/STO) heterostructure possesses a very high mobility (over 500 $cm^2 V^{-1} s^{-1}$ at 2 K). The Ti/Pt layers at the bottom of the substrate serves as a back-gate electrode. Reciprocal space maps (RSM) around the (103) STO peak of this heterostructure shows that the LFO and LAO

films are fully coherent with the STO substrate (Fig. 1b). Additionally, high-angle annular dark field (HAADF) scanning transmission electron microscopy (STEM) image of this heterostructure indicates a cube-on-cube relation between LFO, LAO and STO without the indication of dislocations (Fig. 1c).

## Results

### Gate-controlled resistance response of Broad-spectrum C-2DEG Gas Sensor

The as grown samples exhibit metallic behavior (Fig. 2a)[20]. Due to the introduction of a $LaFeO_3$ delta doping layer, the carrier density of the 2DEG is as low as $2 \times 10^{13}$ /cm². During the cooling or warming, the temperature dependent resistivity curve was smooth without any abnormity. Surprisingly, a back gating at 210 V at 2 K can drive the interface into a fully insulating state with a resistance that is beyond the range of the measurement electronics (10 MΩ). This back-gating driving metal to insulator transition (MIT) is non-volatile and the interface remains insulating if the temperature is kept at 2 K, whose physics will be discussed below. Different from as grown samples where the R-T curve does not exhibit any abnormity, the as-gated insulating interface becomes gas species sensitive, which can be considered to be an activation process for the sensing device. As shown in Fig. 2a, during the warming up process, the interface resistance rather abruptly jumps to a lower value at a specific temperature depending on the gas species in the Quantum Design physical properties measurement system (QD-PPMS) chamber. The temperature of the abrupt resistance change appears to be coincident with the gas sublimation or boiling (SOB) points. For example, when the gas in the chamber is oxygen, the change of the interface resistance occurs around 90 K. If we change the camber gas to be $C_2H_4$, then the resistance change occurs at the SOB temperature of $C_2H_4$.

The sensitivity of this C-2DEG gas sensor in a complex mixed gas environment is one of the key parameters for practical application. To investigate such capability, we mixed a 4.5 Torr multi-gas atmosphere[2] (95% $CO_2$ + 3% $N_2$ + 2% CO) in the PPMS cavity to simulate the Martian atmosphere. After the device was activated at 16.5 K, the warming-up curve (Fig. 2b blue line) shows the SOB points of all three substances. This fact demonstrates that even in a low-pressure environment containing multiple gases, our C-2DEG gas sensor can still sensitively

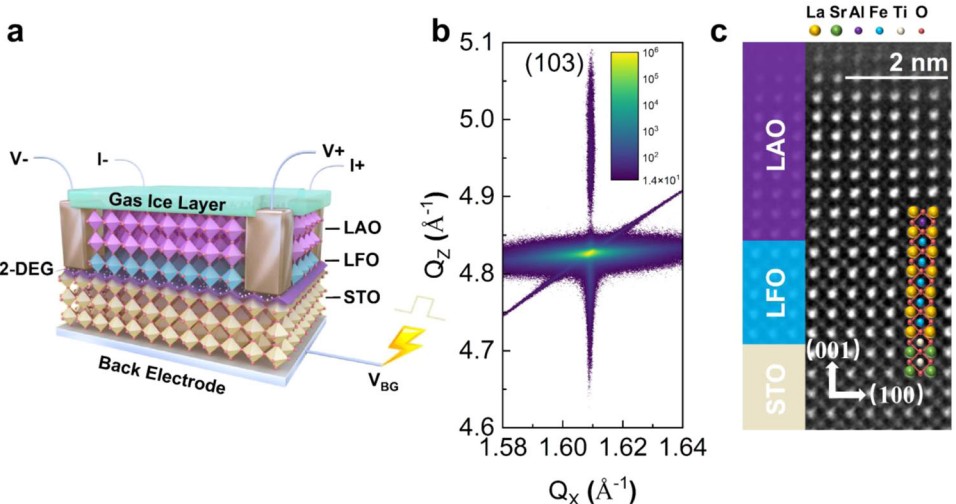

**Fig. 1 | Broad-spectrum correlated two-dimensional electron gas (C-2DEG) Gas Sensor. a**, A schematic of gas senor with back electrode (Pt/Ti) for back-gate. The polyhedra in the diagram are different $ABO_3$ perovskite oxygen octahedra. The A-site atoms are omitted in the schematic diagram, and the vertex corners of the octahedron are all oxygen atoms. Different colors represent different B-site atoms in the octahedral center, from top to bottom, Al (pink), Fe (blue) and Ti (yellow). $V_{BG}$ and the yellow line above represent the pulsed back-gate voltage. **b**, Reciprocal space mapping of (103) peak of the film, indicating that the film is fully coherent with $SrTiO_3$(STO) substrate. The inset color scale represents diffraction intensity in arbitrary units; intensity less than 14 is set as white color in order to remove background noise. **c**, $LaAlO_3$ (LAO, 10-u.c.)/ $LaFeO_3$ (LFO, 5-u.c.)/ $SrTiO_3$ (001) heterostructures. Source data are provided as a Source Data file.

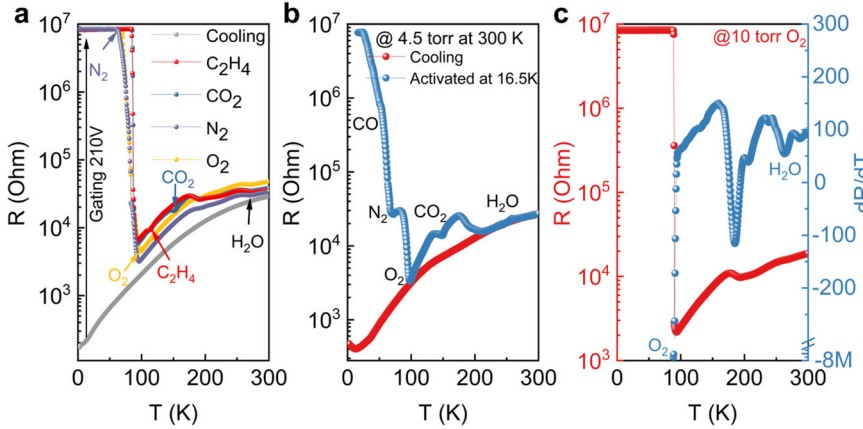

**Fig. 2 | Gate-controlled resistance response of Broad-spectrum C-2DEG Gas Sensor. a** Unique back gating drives non-volatile MIT, resulting in greatly enhanced gas sensitivity in different gas environments, e.g. $N_2$ (purple), $O_2$ (yellow), $C_2H_4$ (red), and $CO_2$ (blue). **b** Responses in a simulated Martian atmospheric environment (95% $CO_2$ + 3% $N_2$ + 2% CO, 4.5 torr measured at 300 K). **c** The curve of resistance (red left axis) and its first order differential (blue right axis) with temperature in 10 torr $O_2$ (measured at 300 K) atmosphere. Source data are provided as a Source Data file.

distinguish different gas components, and can detect gas partial pressure down to 90 mTorr.

## Characteristics of broad-spectrum C-2DEG gas sensor

However, since the SOB of some substances, such as water, just causes a slight change in resistance, which is hard to be observed directly on the resistance versus temperature (R-T) curve. To overcome this challenge, first-order differentiation is used to amplify these minor resistance changes. Furthermore, each valley temperature in the first-order differentiation represents the moment when the substances sublimate or bubble most rapidly into gases. Therefore, the valley temperature corresponds to the critical temperature of sublimation of boiling. To simplify future discussion, this temperature is defined as $T_{SOB}$. The first-order differential of the interfacial resistance under 10 Torr $O_2$ is shown by the blue line in Fig. 2c. It can be seen that the $T_{SOB}$ of oxygen and water are 87 K and 263.5 K, respectively, which are significantly lower than the SOB temperatures at standard atmospheric pressure[21]. This is due to three-phase diagram rule which results in the pressure dependent SOB temperature as we will demonstrate below. Interestingly, our C-2DEG gas detector is capable of detecting small amounts of substances that sublimate or boil at low temperatures, such as $H_2$ (around 17 K)[22], if the device is not activated to an excessively insulating state (Fig. S4a). This feature makes our gas detector tunable based on the SOB temperature of substances to be detected. In addition, it is worth noting that two kinks near 200 K in the first-order differentials are observed in all our C-2DEG gas detectors under different gas environments. These two kinks may result from two new substances produced by the mutual dissolution of gases, such as $CO_2$ and $NO_2$ dissolved in water, which produce carbonic acid (around 180 K) and nitric acid (around 203 K), respectively[23,24]. To confirm that this C-2DEG gas sensor is pressure sensitive, we performed the gating at different pressures and investigate the SOB temperatures. Figure 3a-d show the pressure dependent $T_{SOB}$ of four different gases, $N_2$, $C_2H_4$, $O_2$, and $CO_2$ from 10 Torr to 40 Torr. The $T_{SOB}$ of all substances increase with gas pressure. As shown in Fig. 3e, there is a linear dependence on the $1/T_{SOB}$ of these four substances and the logarithmic value of pressure, which is consistent with the Clausius–Clapeyron relation $\frac{dP}{dT} = \frac{PL}{T^2R}$. Although all gas pressures are measured at 300 K, instead of the temperature when substances sublimate or bubble into gases, the results are still reliable. To further quantitatively determine the measurement limit of the sensor, we analyzed the relationship between the change of the instantaneous conductance slope when substances sublimate or bubble and

the partial pressure of gases. To describe the change in slope, we introduce the phase transition (solid or liquid phase transit into gas phase) angle $\Theta = \tan^{-1}(\frac{k_2}{k_2}) - \tan^{-1}(\frac{k_1}{k_2}) = 45^o - \tan^{-1}(\frac{k_1}{k_2})$ (Fig. 3f)[25]. When $\Theta = 0^o$, the slope of the conductance does not change, indicating that the gas partial pressure is lower than the measurement limit of the device. The larger angle $\Theta$ means a stronger response of resistance change to gas environment. As suggested by the $\Theta$ vs log $P$ plot, the phase transition angle $\Theta$ has a good linear relationship with the logarithmic value of gases partial pressure (Fig. 3g). Therefore, the partial pressure of gases can be further accurately and quantitatively measured by calculating the angle $\Theta$. From the fitting results, it can be deduced that the measurement limits of $CO_2$ and $C_2H_4$ are 0.025 torr and 0.40 torr, respectively (Fig. 3g). If it is in a standard atmospheric pressure environment, they are about 33 ppm and 530 ppm, respectively, which are significantly lower than the measurement limits of previously reported MOS gas sensors (Table S2). Nonetheless, since the resistance at the low-temperature segment exceeds our electronic measurement range, the slope $k_2$ cannot be ascertained, and the diagrams for $N_2$ and $O_2$ are omitted from Fig. 3g.

The change in the interface conductance slope arises from the change in the interfacial carrier concentration, which is entirely determined by the partial pressure of the single gas being measured. According to the aforementioned definition of phase transition angle, the measuring range of a sensor is 0–135°. Under a condition of low partial pressure, devices with low carrier concentrations exhibit larger $k_1$, thereby delivering enhanced resolution for gases possessing low partial pressure. In contrast, as partial pressure escalates, the $|k_1|$ of devices with low carrier concentrations becomes orders of magnitude greater than $|k_2|$. Consequently, $\tan^{-1}(\frac{k_1}{k_2})$ approximates $-90°$, signifying a tendency towards a saturated phase transition angle, thus defines the measurement upper limit of the sensor when using phase transition angle to determine gas partial pressure. In principle, we can extend the upper range by using higher carrier concentrations samples. Therefore, the measurement range of such sensing devices can be rationally designed by controlling the carrier concentrations.

Given that non-polar gases also contribute to the resistance change in the results above, the resistance change does not likely arise from gas absorption. What is more, a back-gating at 210 V for a very short time of only 1 ms at 2 K can also instantly induce non-volatile MIT (Fig. 3h). The broad gas sensitivity together with this fast MIT process at extremely low temperatures further indicates that the back gating induced non-volatile MIT does not arise from any molecular migration

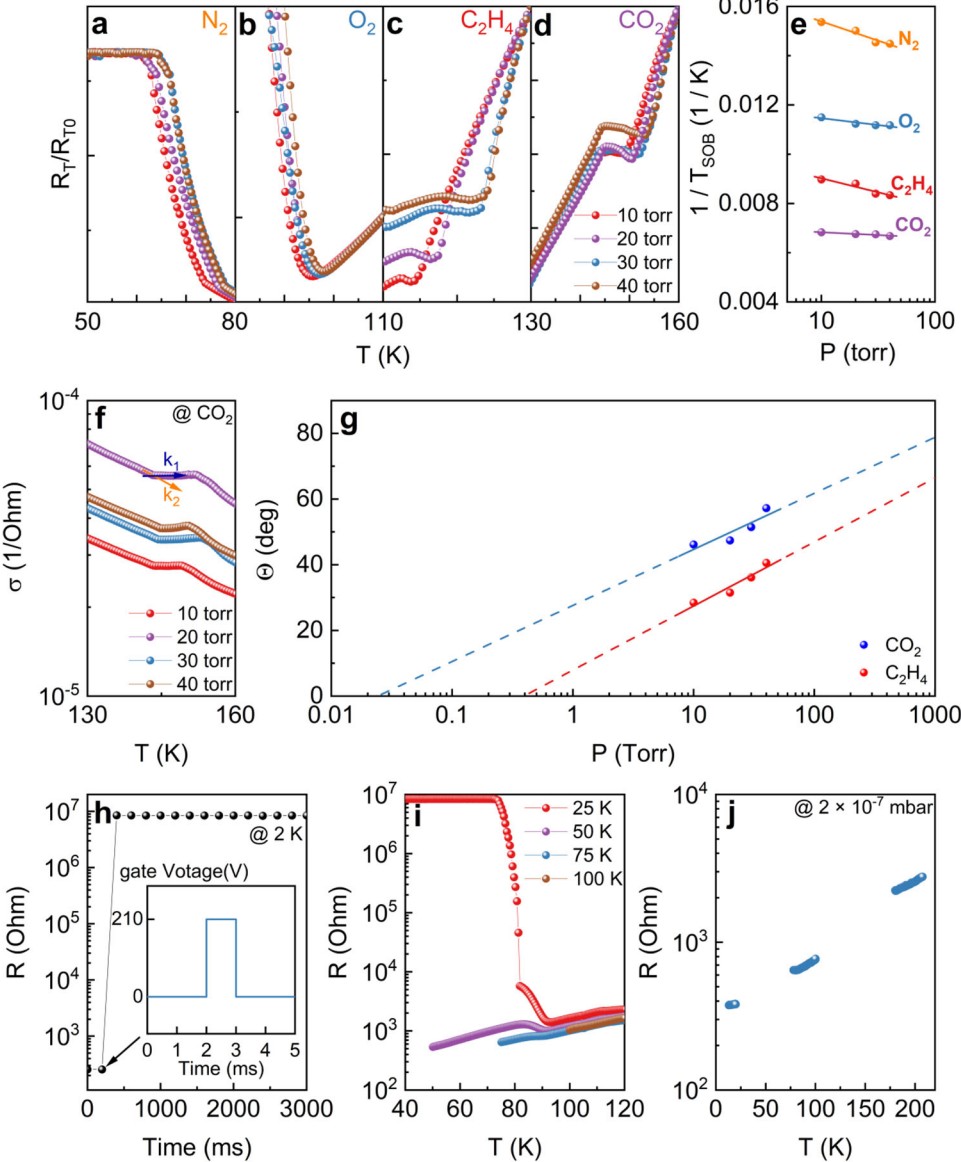

**Fig. 3 | Characteristics of Broad-spectrum C-2DEG Gas Sensor.** Mass spectrum in 10–40 torr (measured at 300 K) $N_2$ **a**, $O_2$ **b**, $C_2H_4$ **c**, and $CO_2$ **d** environments, **b**, **c**, and **d** are normalized at 110 K, 130 K, and 160 K, respectively. **e** $T_{SOB}$ as a function of various gas pressure in an atmosphere of 10–40 torr (measured at 300 K), from top to bottom are $CO_2$, $C_2H_4$, $O_2$, $N_2$. **f** Definition of the phase transition angle Θ. $k_1$ and $k_2$ are the slope of conductance over temperature from below and near $T_{SOB}$. Θ is the angle between $k_2$-normalized $k_1$ and $k_2$ slope lines. **g** Θ as a function of $CO_2$ and $C_2H_4$ pressure. The line is the fitting result. **h** Electrical resistance before and after applying 1 ms 210 V back-gate voltage at 2 K. The inset shows a schematic diagram of the pulsed back-gate voltage applied at Time = 200 ms. **i** *R-T* curves of the device activated before (red and purple) and after (blue and brown) the melting point of oxygen. **j** MIT was not observed in the probe station test in a high vacuum environment ($2 \times 10^{-7}$ mbar). MIT and SOB jumping points of the gas were not observed in the three key temperature regions (13–20 K, 78–100 K, 180–207 K). Source data are provided as a Source Data file.

related process, e.g. gas absorption. Usually, ionic migration, or molecular migration-driven resistance switching is very slow, especially at such low temperature. Taking the ionic gating device as an example, it requires a gating time of several minutes or even hours to drive ion implantation to ensure significant change of resistance. At the same time, if the STO substrate thickness is reduced to about 0.1 mm, a 25 V back gate can also cause the non-volatile MIT, indicating that the electric field plays a central role. Therefore, our observed non-volatile MIT of the millisecond response is more likely to be an electromigration-related process.

Although the above experiments show several examples of activating the gas-sensing performance of 2DEG at extremely low temperature, our research found that the low temperatures are not essential for the device to work. As long as the temperature is below

the solidification temperature of gases, the 2DEG sensor can be activated by the back-gating. Figure 3i shows the warming-up R-T curves at 25–100 K with a temperature interval of 25 K after being activated. Since the triple point of oxygen is 54.36 K[21], oxygen will be completely solidified on the surface of the sensor at 25 K. In contrast, since 50 K is too close to the triple point and there is unavoidable heat exchange on the surface of the C-2DEG sensor, the oxygen is not completely solidified and instead it is in a coexistent solid–liquid phase. At 75 K, all oxygen on the surface should be a liquid state. It can be seen that the change in resistance is more significant when the back-gate electric field is applied at lower temperatures. This observation indicates that the solidified gas has a stronger charging ability and thus can deplete more interface carriers. Therefore, we conclude that a temperature below the freezing point is sufficient to activate the C-2DEG gas sensor,

and thus our C-2DEG gas sensor should be able to operate in a wide temperature range.

Based on the above results, we suspect that the conventional mechanisms of gas adsorption and redox reactions known for MOS gas sensors are no longer applicable to the C-2DEG gas sensor. For starters, it would be impossible for the carriers at the interface to react chemically with the gas molecules on the surface through the insulating layer. Furthermore, the successful detection of non-polar gases and noble gases confirms such issue. Moreover, the non-volatile MIT occurs only below the SOB temperature of the gas and no MIT can be observed in high vacuum ($\sim 2 \times 10^{-7}$ mbar, Fig. 3j). These results demonstrate that the presence of gas ice layer formed by the solidification of gases is essential.

## Discussion

To explain the above observations, a parallel-plate capacitor-like model is proposed to explain the mechanism of this C-2DEG gas sensor. As shown in Fig. 4a, a positive back-gate electric field at low temperature ionizes thin gases and attracts electrons to the gas ice layer. After turning off the back-gate electric field, the bottom electrode will quickly become electrically neutral. While most solidified gases are insulating[26–28], the negative charge will remain in the gas ice layer, creating a negatively charged top gate that rapidly depletes carriers at the 2-DEG interface and induces a non-volatile MIT due to the ultra-thin dielectric layer (about 6 nm). This continuous top gating system is formed by the negatively charged gas ice layer and the 2-DEG electrode. During the warm-up process after back-gating, the SOB of substances will carry away the charges and thus effectively reduce the top gating voltage. As a result, the interface carrier concentration increases. By probing the carrier density at the interface using the Hall effect, a change of carrier concentration occurring at the SOB is directly demonstrated by the temperature dependent carrier concentration as shown Fig. 4b. Around 90 K, corresponding to SOB of

Oxygen, we can see a change of carrier concentration from $5.3 \times 10^{12}$ /cm$^2$ to $8.8 \times 10^{12}$ /cm$^2$. Subsequently, with the SOB of all the substances, the interfacial carrier concentration increases significantly until it returns to the initial value. It is worth mentioning that if applying the back gate at 300 K for two minutes, the cooling and normal warm-up R-T curves are smooth, without the abrupt resistance changes (Fig. S5a). These observations indicate that the gas ice layer is electrically neutral in the normal state, and is only charged after back-gating. The charging ability of different gases may depend on their electron affinity, for example, the C=O bond in $CO_2$ has a stronger ability to bind electrons than the C-H bond in $C_2H_4$, so $CO_2$ has a higher electron affinity (Table S3), which leads to the difference in the measurement limit.

According to the above model of parallel plate capacitors, the depleted interfacial carrier concentration will be equal to the charge concentration of the gas ice layer. To estimate the maximum interfacial carrier concentration that the gas ice layer can deplete, we have grown another heterointerface device at a lower oxygen pressure to achieve a larger a carrier concentration ($2.54 \times 10^{13}$/cm$^2$ for as grown sample at 2 K). After performing an activation process using 210 V for 2 min at 2 K, the resistance of the sensor is found to increase by only an order of magnitude rather than transiting into fully insulating state, indicating that only a fraction of the carriers was depleted. It can be seen from Fig. S6 that the carrier mobility is constant at same temperature, so the depleted carrier concentration then can be estimated to be about $2.24 \times 10^{13}$/cm$^2$, which thus is almost the same as the gas charge concentration we calculated above. Accordingly, a sample with carrier concentration lower than $\sim 2.24 \times 10^{13}$/cm$^2$ is more easily to be driven into fully insulator state by back gating. In addition, the depleted carrier concentration can be expressed as $n = \varepsilon_r \varepsilon_0 V/d = \varepsilon_r \varepsilon_0 E$. Here, $\varepsilon_r$ is the relative dielectric constant, $\varepsilon_0$ is the vacuum dielectric constant and $E$ is the electric field intensity. Utilizing this model, the relative dielectric constant of our STO substrate is estimated to be 9600 at 2 K,

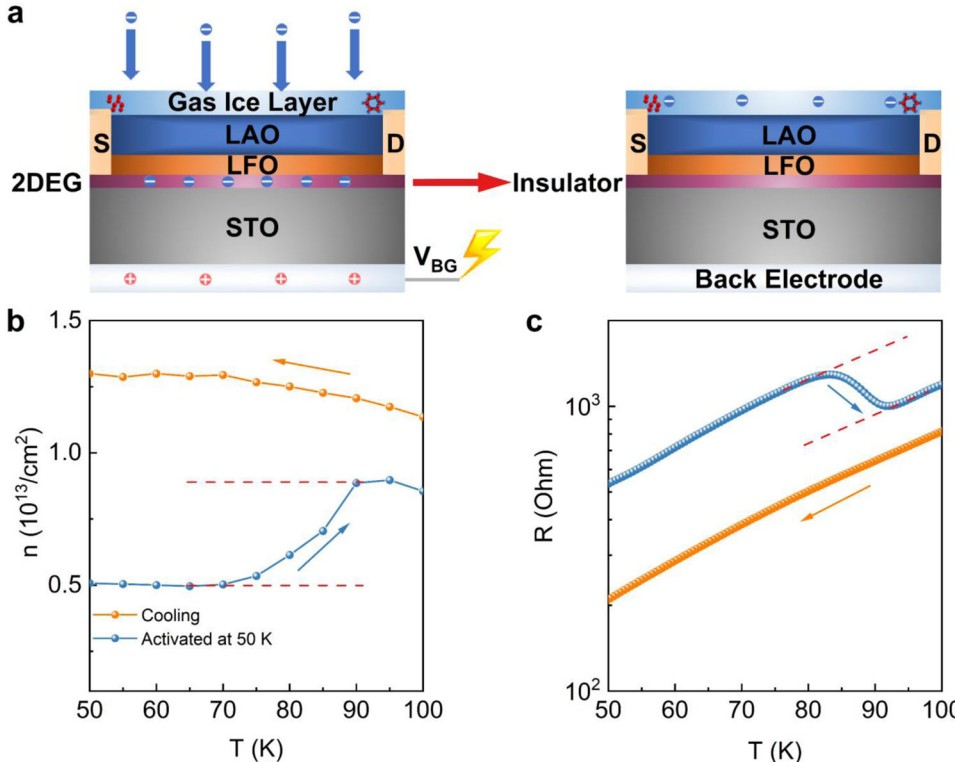

**Fig. 4 | Mechanism of broad-spectrum C-2DEG Gas Sensor. a** Schematic diagram of depletion of interface carriers due to charging of gas ice layer. The molecular structures of solid oxygen (left) and ice (right) are shown in the gas ice layer. The experimentally observed change in carrier concentration n **b** after activation at 50 K (blue curve) resulted in the change of the slope of the electrical transport curve **c**. Source data are provided as a Source Data file.

which is close to actually $\varepsilon_r$ of single crystal STO[29,30], further supporting our proposed model of parallel plate capacitors. Based on the underline physics for the gas sensing of our 2DEG system, we can anticipate similar outcomes for other 2DEG systems, such as 2DEG in III-V semiconductors. However, it is worth noting that the relative dielectric constant of single crystal $SrTiO_3$ at low temperatures ($\leq 25$ K) is three orders of magnitude larger than III-V materials like GaAs and GaN[29–31]. Consequently, to achieve the same depleted carrier concentration, AlGaAs/GaAs and AlGaN/GaN systems necessitate an electric field ($E$) that is three orders of magnitude more intense than that required for the $LaAlO_3/SrTiO_3$ system.

It is worth noting that the sublimation jumping point of methane and carbon dioxide can still be observed in the 50% $CH_4$ + 50% $CO_2$ mixed gas environment (Fig. S5b) using the device with higher carrier concentration ($n_s = 2.54 \times 10^{13}$/cm$^2$ at 2 K). However, its resistance response to $CO_2$ is much weaker than that of samples with low carrier concentration. For example, the phase transition angle $\Theta = 1.08°$ for this sample is much smaller than that (46.16°, Fig. 3g) for the low carrier concentration sensor with same $CO_2$ partial pressure. This fact indicates that lower interfacial carrier concentrations can lead to a larger resistive response to the same gas at the same partial pressure. Introduction of delta doping of LAO/STO interface with a $LaFeO_3$ layer can significantly reduce the carrier concentration, and as a result, can essentially enhance the gas sensitivity.

The mechanism of non-volatile MIT induced by the back-gate at low temperature supports that 2-DEG at the LAO/LFO/STO interface has broad-spectrum gas sensitivity, which can accurately and quantitatively measure gases partial pressure and detect a broad spectrum of gases, including inert gas (e.g. $N_2$), rare gas (e.g., Ar), polar gas (e.g. $CH_4$) or non-polar gas. Its measurement limit can be expected to be the mTorr level or even lower. Especially the measurement limit of $CO_2$ is equivalent to 33 ppm at a standard atmospheric pressure, comparable to the best MOS gas sensor reported (Table S2). In addition, this back-gate induced non-volatile MIT is an extremely fast or even ultrafast process, which is of great significance to the practical application of the device, and we will continue to explore its response speed in future.

Given that the nighttime temperature of many planets is very low, the C-2DEG should have application potential as a low cost and high-efficient sensor for planetary gas detection. For example, the lowest temperature at the poles of Mars can reach about 130 K, which is lower than the solidification temperature of $CO_2$. Therefore, the gas detection capability can be obtained by activating the C-2DEG gas sensor through a back-gate. Since a simple multi-stage thermoelectric refrigeration[32] can low down the temperature by over 60 K, the working temperature range of a C-2DEG equipped with thermoelectric refrigeration can be significantly enlarged and thus can routinely work for a wide range of gas species, such as $N_2$, $CH_4$, and $O_2$, etc.

## Methods

### Sample growth
$LaAlO_3/LaFeO_3/SrTiO_3$ samples are epitaxially grown on (001)-oriented $SrTiO_3$ substrate by pulsed laser deposition (PLD, KrF, $\lambda = 248$ nm). The substrate is $TiO_2$ terminated by reacting in deionized water for 30 min and etching in BOE for 30 s. Then the $SrTiO_3$ substrate is annealed at 950 °C for 90 min to achieve an atomically smooth surface. Both $LaAlO_3$ thin film and $LaFeO_3$ thin film are subsequently deposited on the top of $TiO_2$-terminated SrTiO3 (001) substrate by home-made PLD at a temperature of 700 °C with the laser energy density of 1 to 1.25 J/cm$^2$, and $2 \times 10^{-5}$ to $1 \times 10^{-4}$ mbar oxygen pressure, monitored by reflection high-energy electron diffraction (RHEED) during the growth, and then are gradually cooled down to room temperature. The adhesion layer Ti and the bottom electrode Pt at the bottom of the substrate are both deposited by magnetron sputtering.

### Structural characterization and STEM Imaging
The crystal structure and reciprocal space mapping (RSM) were characterized by an X-ray Diffractometer (Panalytical Empyrean Alpha 1). Atomic force microscopy (AFM) images were performed on the Oxford Cypher (contact mode, AC240TSA-R3 tip). X-ray absorption spectroscopy (XAS) was conducted at the Hefei Light Source (National Synchrotron Radiation Laboratory). Fe L2,3-edge were detected with incident light normal to the sample surface. High-angle annular dark field scanning transmission electron microscopy (HAADF-STEM) was performed on an aberration-corrected FEI Titan Themis G2 operated at 300 kV at the Institute of Physics, Chinese Academy of Science.

### Transport measurement
Transport measurements under gases atmosphere were performed in a PPMS setup (Quantum Design DynaCool system). Vander Pauw method was employed for all electrical transport tests. Wires and gases were introduced by a homemade multifunction probe. The resistance and pulse back-gate voltage were recorded and output by the digital source meter Keithley 2614 B and Keithley 2400 respectively, both controlled by home-made labview programs.

High-vacuum transportation measurements were performed in a cryogenic probe station (Lakeshore CRX-VF) at Nanchang University. The back-gate voltage and resistance were output and read by the PDA FS-pro and a digital lock-in amplifier (Zurich Instruments MFLI, 0.5 μA constant current mode). Due to the inconsistent thermal expansion coefficients of the sample stage and the probe, poor contact is prone to occur during the heating process. Therefore, measurements were limited to three key temperature ranges (13–20 K, 78–100 K, 180–207 K).

## Data availability
The data that support the findings of this study are available from the corresponding author upon request. Source data are provided in this paper.

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

## Acknowledgements

We thank Guanglei Cheng for the invaluable discussion. This work was partially carried out at the MESA+ Institute for Nanotechnology and the USTC Center for Micro and Nanoscale Research and Fabrication. This work was supported by the National Key R&D Program of China (No. 2022YFA1403000 (Z.L.)) and the National Natural Science Foundation of China (Nos. 52272095 (Z.L.) and 11974325 (Z.L.)).

## Author contributions

Z.L. and G.R. directed the project. Y.H. did most of the design and fabrication of the devices, performed the experiments, and wrote the manuscript. Z.L. supervised sample growth and reviewed the manuscript. Q.Z. performed the measurements of HAADF-STEM. X.L. and Y.Z. contributed to high-vacuum transportation measurements. L.Wei, Z.D., T.L., J.L., W.X., S.H., L.Wang, L.L., and Mark Huijben contributed to measurements and data analysis. L.Wang contributed to the manuscript writing. Y.G., K.C., G.K., and G.R. reviewed the manuscript.

## Competing interests

The authors declare no competing interests.
