## [Peer Review File · Nature Communications]

REVIEWER COMMENTS

Reviewer #1 (Remarks to the Author):

This manuscript presented a study titled "A Broad-spectrum Gas Sensor based on Correlated Two-dimensional Electron Gas." In their study, the authors demonstrated a broad-spectrum gas sensor utilizing correlated two-dimensional gas (C-2DEG) at a delta-doped LaAlO₃/SrTiO₃ interface with LaFeO₃, metal-insulator-transition (MIT) in the presence of gas species using back-gating on the 2DEG at lower temperatures, and sublimation and boiling of gases (SOB).

This method of achieving a broad-spectrum gas sensor is unique and novel. This technique provides a way to measure all types of gases—which is remarkable. However, the authors need to address the comments below to address the versatility and usability of the proposed broad-spectrum gas sensor.

1. Based on Fig 3(e), particularly when looking at N₂ and O₂ (Fig. 3(a), (b), and (e)), 1/T SOB seems to be saturated with increased partial pressure. Please suggest if this is the case. Is it the measurement limitation due to a particular concentration of 2DEG? With this limitation, can the gas partial pressure be measured accurately and quantitatively? Determining the sensor response as a function of gas partial pressure is extremely important—at least, the limitation of the current sensor may need to be informed.
2. Continuing from question 1, On page 7, line 160, the authors stated that the interface conductance slope entirely depends on the partial pressure of a particular gas measured. On page 10, line 247, the authors indicated that lower 2DEG concentration has a higher response and more prominent slope change. These two statements combined suggest that for a particular partial pressure of the gas (assume higher pressure) and with higher 2DEG, the θ can be extremely small, leading to lower resolution in determining the partial pressure of the gas. Please comment on this.
3. For continuation, the authors are requested to place N₂ and O₂ graphs in Fig. 3(g). If not placed, please mention in the manuscript why the data are not considered for Fig. 3(g).
4. The authors explained non-volatile MIT in the presence of back gating and at lower temperatures. In this process, the charge depletion of 2DEG depends on the ionization of gases above the gas ice layer and the trapping of electrons in the gas ice layer. The number of electrons getting trapped in the gas ice layer and the depletion of the 2DEG depends on the ionization gas environment (probably also depending on the pressure of the measurement chamber) and the concentration of 2DEG. Do authors suggest a particular 2DEG concentration of their device that can work with different gas environments and even with lower and higher pressures of ionization gas? Or does the measurement chamber of these devices need to have certain gas pressure, which has to be controlled? Please address how to control the density of electrons getting trapped in the ice layer in the manuscript.
5. As discussed in the manuscript, for better performance of this sensor, a lower 2DEG concentration is sufficient. In this case, what are the advantages of using the LaAlO₃/SrTiO₃ system compared to AlGaAs/GaAs and AlGaIn/GaN systems? The same concept used in AlGaAs/GaAs and AlGaIn/GaN systems may even lead to better performances due to the higher mobility of 2DEG. Moreover, the mentioned systems are optimized, and sensors can be fabricated on large-diameter wafers leading to better performance at lower cost. Please comment on this. Please mention in the manuscript why LaAlO₃/SrTiO₃ is superior compared to other similar 2DEGs systems.
6. In the abstract (page 2, line 30), correlated two-dimensional gas (C-2DEG) has to be replaced with correlated two-dimensional electron gas (C-2DEG). Please modify.

I thank the authors for going through above comments.

Best regards,
Ravikiran,
Senior research scientist, NTU, Singapore.

Reviewer #2 (Remarks to the Author):

This manuscript reports a broad-spectrum gas sensor utilizing correlated two-dimensional gas (C-2DEG) at a delta-doped LaAlO₃/SrTiO₃ interface with LaFeO₃. Research on the gas sensing is viable topic to the field, however, this paper does not provide new sensor concepts or sufficient sensor performance. Overall, this paper is not suitable for Nature Communications, which requires novel ideas.

1. How low is the contact resistance at low temperature? Because of adhesion issue with Ti/Pt at low temperature, the contact resistance increase.
2. Sensor must response selectively for specific gas while the proposed sensor does not show a good selectivity for target gas.

Reviewer #3 (Remarks to the Author):

In this study, the authors designed novel two-dimensional gas (C-2DEG) at a delta-doped LaAlO₃/SrTiO₃ interface with LaFeO₃ to fabricate broad-spectrum gas sensor, which can sensitively and quantitatively probe very broad gas species. The content is interesting and valuable, and related experiments are abundant and discussed in detail. However, the article is poorly organized and less innovative, and some scientific problems should be fully considered.

1. In the "Abstract", The features and highlights of the article are not reflected, especially the unique performance of the sensor.
2. In the "Introduction", the research bottlenecks of the broad-spectrum gas sensor should be discussed, especially for the role of high-mobility two-dimensional electron gas.
3. Why did you choose this material? LaAlO₃/SrTiO₃, how about other similar materials?
4. In order to evaluate the structure of two-dimensional gas (C-2DEG) at a delta-doped LaAlO₃/SrTiO₃ interface with LaFeO₃, and more structural characterization should be considered, such as XPS, XRD, FTIR and so on.
5. Although Figure 4 exhibited the possible mechanism of broad-spectrum C-2DEG gas sensor, the deeper mechanism still needs to be discussed again, and especially for the effect of temperatures.
6. In order to evaluate the advantages of this work, similar studies should be compared and discussed.

REVIEWER COMMENTS

We are very pleased to see that both reviewers gave us positive comments on the manuscript. We sincerely thank the reviewers for their constructive suggestions to further improve the manuscript. Below, we reply point by point to all comments and questions raised by the reviewers and note the corresponding changes in the revised manuscript.

Response to reviewers' concerns:

For Reviewer#1 (Remarks to the Author):

This manuscript presented a study titled “A Broad-spectrum Gas Sensor based on Correlated Two-dimensional Electron Gas.” In their study, the authors demonstrated a broad-spectrum gas sensor utilizing correlated two-dimensional gas (C-2DEG) at a delta-doped LaAlO₃/SrTiO₃ interface with LaFeO₃, metal-insulator-transition (MIT) in the presence of gas species using back-gating on the 2DEG at lower temperatures, and sublimation and boiling of gases (SOB).

This method of achieving a broad-spectrum gas sensor is unique and novel. This technique provides a way to measure all types of gases—which is remarkable. However, the authors need to address the comments below to address the versatility and usability of the proposed broad-spectrum gas sensor.

Our response: We sincerely appreciate the reviewer for his high evaluation of our work. In the following, we clarify/address the reviewer's concerns and accordingly indicate the changes made in the revised manuscript and Supplementary Information (SI).

Comment 1: Based on Fig 3(e), particularly when looking at N₂ and O₂ (Fig. 3(a), (b), and (e)), $1/T_{\text{SOB}}$ seems to be saturated with increased partial pressure. Please suggest if this is the case. Is it the measurement limitation due to a particular concentration of 2DEG? With this limitation, can the gas partial pressure be measured accurately and quantitatively? Determining the sensor response as a function of gas partial pressure is extremely important—at least, the limitation of the current sensor may need to be informed.

Our response: According to the Clausius-Clapeyron relationship ($1/T$ is proportional to $\ln P$), the change of T_{SOB} will gradually decrease with increasing the gas partial pressure. For example, when pressure increases from 30 torr to 40 torr, the change of T_{SOB} is less than 0.5 K. In our experiment, the resistivity was measured with a temperature step of 0.5 K, so at high pressure, the change of temperature is too small to be extracted from our R-T curve. Indeed, as the reviewer pointed out, the $1/T_{\text{SOB}}$ will saturate eventually due to limited test accuracy. Regarding the impact of concentration of 2DEG, it actually will not play a strong role in the shift of the temperature with pressure, since the T_{SOB} just follows Clausius-Clapeyron relationship. In our experiment, we did show same T_{SOB} for different carrier concentration, as shown in Fig. R1. In addition, the actual application scenarios are mostly mixed gas environments. The pressure P in the Clausius-Clapeyron relationship refers to the total gas pressure, not the partial pressure of a single gas. The Clausius-Clapeyron relation is not suitable for calculating partial pressures in mixed gas environments. However, the T_{SOB} from fundamental side can provide us a strong evidence to understand the mechanism of C-2DEG sensor, though limitation still exists in the application side.

Therefore, we propose a phase transition angle to more accurately calculate partial pressure of different gases in a mixed gas environment. Using phase transition angle, we can determine gas partial pressure more precisely over a wider range (see Figure 3f, 3g in the main text, and page 6 & 7). The effect of carrier concentration on phase transition angle will be discussed in the response to comment 2.

Figure R1 | Temperature response of sensors with different carrier concentrations in 20 torr C₂H₄. The circle represents the sensor with a carrier concentration of $1.30 \times 10^{13} / \text{cm}^2$, and the square represents the sensor with a carrier concentration of $2.54 \times 10^{13} / \text{cm}^2$. The difference between the inflection point of the resistance and the minimum value of the first-order differential of the resistance is less than 0.5 K within the error bars.

Comment 2: Continuing from question 1, On page 7, line 160, the authors stated that the interface conductance slope entirely depends on the partial pressure of a particular gas measured. On page 10, line 247, the authors indicated that lower 2DEG concentration has a higher response and more prominent slope change. These two statements combined suggest that for a particular partial pressure of the gas (assume higher pressure) and with higher 2DEG, the θ can be extremely small, leading to lower resolution in determining the partial pressure of the gas. Please comment on this.

Our response: According to Fig. 3g in the main text, for the same device (sample) under identical activation conditions, the change of the interface conductance slope depends on the partial pressure of the tested gas. Specifically, the greater the gas partial pressure, the larger the change of the interface conductance slope. Meanwhile, if both the gas and partial pressure are certain, the change of carrier concentration at interface is the same for devices (samples) with different carrier concentrations during the

sublimes or boils. Therefore, the devices with higher interfacial carrier concentration would show a smaller change of the conductivity slope than the ones with lower interface carrier concentration, due to the small ratio of $n(\text{changed})/n(\text{original})$. Simultaneously, according to the definition of phase transition angle, the measuring range of a sensor is $0 - 135^\circ$. Under low partial pressure, devices with lower carrier concentrations exhibit larger k_1 , thus offering enhanced resolution for gases with low partial pressure. In contrast, as partial pressure escalates, the magnitude of $|k_1|$ becomes orders of magnitude greater than $|k_2|$ in the devices with low carrier concentrations. Consequently, $\tan^{-1}\left(\frac{k_1}{k_2}\right)$ approximates -90° , signifying a tendency towards a saturated phase transition angle, thus it defines the measurement upper limit of the sensor when using phase transition angle to determine gas partial pressure. In principle, we can raise the upper limit by using higher carrier concentration samples. Therefore, the measurement range of such sensing devices can be rationally designed by controlling the carrier concentration.

On page 11, the following sentence “This fact indicates that lower interfacial carrier concentrations can have a stronger gas response.”

has been changed into “This fact indicates that lower interfacial carrier concentrations can lead to a larger resistive response to the same gas at the same partial pressure.”

Comment 3: For continuation, the authors are requested to place N_2 and O_2 graphs in Fig. 3(g). If not placed, please mention in the manuscript why the data are not considered for Fig. 3(g).

Our response: Since the resistance at the low-temperature segment exceeds our electronic measurement range, the slope k_2 cannot be ascertained, and the diagrams for N_2 and O_2 are omitted from Fig. 3g. In reality, the sensor is extremely sensitive to oxygen, leading to the k_2 of $\sim 10^{-7}$, while the k_1 corresponding to oxygen of $\sim 10^{-4}$, where the phase transition angle has become saturated. As mentioned in the reply to comment 2, although devices with low carrier concentrations can recognize O_2 of high partial pressure, it is difficult to accurately quantify the partial pressures. Indeed, for an

accurate quantitative assessment of the partial pressure of O₂ at high levels, it is recommended to utilize sensors with higher carrier concentrations. This strategy effectively extends the upper limit of the measurement range, ensuring precise measurements even at high O₂ partial pressure.

On page 7, the following sentence “Nonetheless, since the resistance at the low-temperature segment exceeds our electronic measurement range, the slope k_2 cannot be ascertained, the diagrams for N₂ and O₂ are omitted from Fig. 3g.” is added.

Comment 4: The authors explained non-volatile MIT in the presence of back gating and at lower temperatures. In this process, the charge depletion of 2DEG depends on the ionization of gases above the gas ice layer and the trapping of electrons in the gas ice layer. The number of electrons getting trapped in the gas ice layer and the depletion of the 2DEG depends on the ionization gas environment (probably also depending on the pressure of the measurement chamber) and the concentration of 2DEG. Do authors suggest a particular 2DEG concentration of their device that can work with different gas environments and even with lower and higher pressures of ionization gas? Or does the measurement chamber of these devices need to have certain gas pressure, which has to be controlled? Please address how to control the density of electrons getting trapped in the ice layer in the manuscript.

Our response: Thank you very much for the professional comments. Indeed, charge depletion of 2DEG strongly depends on the gas environment and pressure. With lower pressure, the number of molecules solidified on the surface of the film is smaller and the number of electrons that each molecule can accommodate is limited. Therefore, the chamber pressure will not only affect the phase transition angle as demonstrated in Fig. 3g in the main text, but also affect the activation process. For example, in the high vacuum, the 2DEG cannot be switched to the insulating state at high vacuum experiment at low temperature (see Fig. 3j in the main text), and this device also does not show any sensing behavior. Therefore, there must be a minimum pressure to form a gas ice layer on the surface of the film to activate the sensor. However, higher pressure will not hinder the operation of the sensor. At low temperatures, most gaseous

substances become solid or liquid state. At this point, the actual pressure will be so low that the thin gas is sufficiently ionized to activate the sensor.

Actually, activating the device at different pressure and gas environments has been tried (see Fig. 2 in the main text), and we can observe the sensor performing well under different conditions. Regarding the control of trapped electrons in the ice layer, there are different ways. According to the capacitor model $Q = CV$, the maximum number of electrons captured by the gas ice layer depends entirely on the gating voltage. Therefore, when the number of molecules in the gas ice layer is large enough, changing the gating voltage is the most direct way to control the trapped electrons in the ice layer. As shown in Fig. R2, we could change the non-volatile resistance switching ratio by using different gating voltages, which means different numbers of trapped electrons. In addition, if the gas pressure is so low that there are not enough molecules in the gas ice to accommodate the aforementioned number of electrons, the number of electrons trapped in the gas ice layer can also be limited by the gas pressure. Therefore, one can also tune the trapped electrons by tuning the gas pressure.

Figure R2 | Different V_{BG} to control resistance switching ratios, thus equivalent to the trapped electrons concentration by the gas ice layer.

As the reviewer mentioned, the performance of this device also strongly depends on the carrier concentration since the trapped electron in the gas ice layer is limited according

to the parallel-plate capacitor series model. Samples with too high carrier concentration typically exhibit very weak resistance change at the gas SOB temperature (Fig. S5b in the SI).

As the Fig. 3 shown in main text, as long as the temperature remains below the SOB temperature of gases, the 2DEG sensor can be activated by back-gating and subsequently exhibits sensing capability. Therefore, the device can work in different gas environments, making it a promising sensor for a broad-spectrum of gases (see Fig. 3 in the main text).

In a word, there are several ways to tune the device's performance according to the technique requirement (e.g., carrier concentration of as-grown samples, gating voltage and gating back-ground pressure). These facts also suggest such devices hold great promise for further engineering for practical application.

It is also worth mentioning that the electronic state of the C-2DEG can be easily controlled by the electric field to realize normally-on transistor devices with different resistance switching ratios. What is more, a low-temperature transistor for writing and erasing has been achieved through the coordinated control of the optical field and the electric field. Relevant data will be reported in the next paper, with a portion of the data shown in Fig. R3.

Figure R3 | Different voltages to achieve different non-volatile MIT with different

resistance switch ratios.

Comment 5: As discussed in the manuscript, for better performance of this sensor, a lower 2DEG concentration is sufficient. In this case, what are the advantages of using the LaAlO₃/SrTiO₃ system compared to AlGaAs/GaAs and AlGaN/GaN systems? The same concept used in AlGaAs/GaAs and AlGaN/GaN systems may even lead to better performances due to the higher mobility of 2DEG. Moreover, the mentioned systems are optimized, and sensors can be fabricated on large-diameter wafers leading to better performance at lower cost. Please comment on this. Please mention in the manuscript why LaAlO₃/SrTiO₃ is superior compared to other similar 2DEGs systems.

Our response: LaAlO₃/SrTiO₃ has a strong correlation effect, and the two-dimensional electron gas can be controlled simply by an electric field, which shows the prospect of developing oxide electronic devices, as discussed in the introduction of our manuscript. Further, the high dielectric constant of STO makes it ideal for gating experiments. According to the parallel plate capacitor model, the depleted carrier concentration $n = Q/S = CV/S = \epsilon_r \epsilon_0 V/d = \epsilon_r \epsilon_0 E$. Where Q is the amount of charge in the gas ice layer, S is the electrode area, C is the capacitance, ϵ_r is the relative dielectric constant, ϵ_0 is the vacuum dielectric constant and E is the electric field intensity. The very high dielectric constant of STO thus allows us to tune the carriers over a very wide range.

In addition, the lower the carrier concentration, the better the device performance for low partial pressure according to our model. Carrier concentration can be well controlled in LAO/STO system by either growth condition (e.g., oxygen partial during growth [S. Thiel *et al.*, *Science* **313**, 1942 (2006)] or delta doping [T. Fix *et al.*, *Applied Physics Letters* **94**, 172101 (2009)]). With LFO delta doping, we indeed realized very low carrier concentration. All these features make LAO/STO a very attractive system for device applications, including gas sensor.

Inspired by the reviewer, we believe that this mechanism is also feasible for other similar 2DEG systems. Although we have not currently performed studies on III-V materials which are not within our research area, it is worth to perform further research

on III-V semiconductors among this community. In addition, it has been reported that the relative dielectric constant of SrTiO₃ at low temperature (≤ 25 K) is three orders of magnitude larger than III-V materials such as GaAs and GaN [I. Strzalkowski *et al.*, *Applied Physics Letters* **28**, 350 (2008)]. If the same carrier concentration is depleted, AlGaAs/GaAs and AlGaN/GaN systems would require an electric field (E) that is three orders of magnitude more intense than that needed for the LaAlO₃/SrTiO₃ system. In the other way, this situation also suggests that other 2DEG systems with low dielectric constant would need to have lower carrier concentration in order to exhibit very good performance.

In addition, using this model, the relative dielectric constant of our STO substrate is calculated to be 9600 at 2 K, which is close to the actual ϵ_r of single crystal STO [O. G. Vendik *et al.*, *Journal of Superconductivity* **12**, 325 (1999) & M. J. Coak *et al.*, *Physical Review B* **100**, 214111 (2019)]. This fact also further validates our proposed model. In order to demonstrate the superiority of LaAlO₃/SrTiO₃ compared to other similar 2DEGs systems, on pages 10 & 11, the following paragraph “In addition, the depleted carrier concentration can be expressed as $n = \epsilon_r \epsilon_0 V/d = \epsilon_r \epsilon_0 E \dots$. Consequently, to achieve the same depleted carrier concentration, AlGaAs/GaAs and AlGaN/GaN systems necessitate an electric field (E) that is three orders of magnitude more intense than that required for the LaAlO₃/SrTiO₃ system.” is added.

Comment 6: In the abstract (page 2, line 30), correlated two-dimensional gas (C-2DEG) has to be replaced with correlated two-dimensional electron gas (C-2DEG). Please modify.

Our response: We apologize for the mistake in the original manuscript. Taking the reviewer’s suggestion, we have corrected it in the manuscript.

For Reviewer#2 (Remarks to the Author):

Comments: This manuscript reports a broad-spectrum gas sensor utilizing correlated two-dimensional gas (C-2DEG) at a delta-doped $\text{LaAlO}_3/\text{SrTiO}_3$ interface with LaFeO_3 . Research on the gas sensing is viable topic to the field, however, this paper does not provide new sensor concepts or sufficient sensor performance. Overall, this paper is not suitable for Nature Communications, which requires novel ideas.

Our response: We appreciate the reviewer's acknowledgment of our delta-doped $\text{LaAlO}_3/\text{SrTiO}_3$ interface with LaFeO_3 for gas sensing, but do not fully agree with other comments regarding the novelty or sufficient sensor performance. Here, we would like to take this opportunity to reiterate the significance of our achievement in this fundamentally new gas sensor based on correlated two-dimensional electron gas (C-2DEG). Our work demonstrated a truly functional C-2DEG device driven by back-gating, and proposes a novel purely physical gas sensing mechanism. Our device does not require chemical reactions with the gases, eliminating the limitations from operating temperature, response time, and susceptibility to cycle. This makes it ideal for extreme environments. We have successfully developed a cost-effective and structurally simple gas sensor based on a delta-doped correlated two-dimensional electron gas (C-2DEG) ($\text{LaAlO}_3/\text{LaFeO}_3/\text{SrTiO}_3$). First of all, our research reveals that back-gating of this C-2DEG can rapidly induce a non-volatile metal-insulating transition. As a consequence, the typically trivial C-2DEG exhibits a remarkable sensitivity to a wide range of gas species, no matter polar gas, nonpolar gas, inert gas, or rare gas, such as hydrogen, nitrogen, methane, oxygen, ethylene, carbon dioxide, water, and gasified acids, among others. Unlike traditional metal oxide semiconductor (MOS) gas sensors, the working mechanism of our C-2DEG gas sensor is fundamentally distinct. This unique characteristic ensures that it remains unaffected by mixed gases and enables operation across a wide temperature range. This characteristic makes it particularly well-suited for environments with extreme temperature variations, such as planetary exploration. What is more, our C-2DEG gas sensor also plays sensitivity to gas partial pressure, and the change of resistance slope follows a simple linear function with the logarithm of

gas partial pressure. In addition, in terms of device performance, the measurement limit for CO₂ surpasses that of currently reported metal oxide sensors.

Comment 1: How low is the contact resistance at low temperature? Because of adhesion issue with Ti/Pt at low temperature, the contact resistance increase.

Our response: We agree that contact resistance may increase due to adhesion issue with Ti/Pt at low temperature. However, in our experiment, the resistivity was measured using Van der Pauw method. It is well known that the four probe Van der Pauw configuration can measure the resistivity of a sample without any impact from contact resistance. Therefore, the adhesion issue of electrode can be ignored. In addition, Ti/Pt is only used as the bottom metal electrode of the parallel plate capacitor, and its contact resistance with the SrTiO₃ substrate has not affect the parallel plate capacitor. In addition, the SrTiO₃ substrate as a dielectric in the middle is also highly insulating, and the energy band gap is about 3.2 eV.

Comment 2: Sensor must response selectively for specific gas while the proposed sensor does not show a good selectivity for target gas.

Our response: We do not fully agree with the reviewers. Part of the commercial gas sensor is to select a specific gas, such as monitoring gas leaks, etc. In addition, there are some sensors which can select multiple gases, and many occasions require multiple gas detection, such as geological science, planetary exploration, etc. As mentioned in the introduction, the development and improvement of broad-spectrum gas sensors are ongoing in current technology, aiming to reduce costs and enhance sensitivity to different types of gases. Our C-2DEG gas sensor possesses broad-spectrum capabilities. Of course, it is technically possible to achieve detect some specific gases with our developed C-2DEG sensor, such as selectively measuring a certain type of gas through gas filtration to achieve single gas selection. These capabilities can be realized at the technical level.

For Reviewer#3 (Remarks to the Author):

In this study, the authors designed novel two-dimensional gas (C-2DEG) at a delta-doped $\text{LaAlO}_3/\text{SrTiO}_3$ interface with LaFeO_3 to fabricate broad-spectrum gas sensor, which can sensitively and quantitatively probe very broad gas species. The content is interesting and valuable, and related experiments are abundant and discussed in detail. However, the article is poorly organized and less innovative, and some scientific problems should be fully considered.

Our response: We thank the reviewer for considering that our manuscript is interesting and valuable, and the constructive comments which can greatly help us improve our manuscript. Below we will address all raised comments point by point.

Comment 1: In the “Abstract”, The features and highlights of the article are not reflected, especially the unique performance of the sensor.

Our response: We highly appreciate this helpful suggestion. To address the reviewer’s comment, we have rewritten the abstract to

“Designing a broad-spectrum gas sensor capable of identifying gas components in complex environments, such as mixed atmospheres or extreme temperatures, is a significant concern for various technologies, including energy, geological science, and Furthermore, its readout is a simple measurement of electric resistance change, thus providing a very low-cost and highly efficient broad-spectrum sensing technique.”

2. In the “Introduction”, the research bottlenecks of the broad-spectrum gas sensor should be discussed, especially for the role of high-mobility two-dimensional electron gas.

Our response: We sincerely thank the reviewer’s suggestion. We have added a description of the research bottlenecks of the broad-spectrum gas sensor and the role of high-mobility two-dimensional electron gas.

On page 3, the following three sentences “On the other hand, numerous metal-oxide-semiconductor (MOS) gas sensors exhibit cross-responses and limited selectivity of

gases, and also typically only work for a few specific gases after doping, light or temperature modulation.”

“Furthermore, owing to the inherent stable chemical characteristics of inert and rare gases, their selection through conventional methods is very challenging. This situation further restricts the development of truly broad-spectrum gas sensor.”

“The electronic state of the C-2DEG can be easily controlled by an electric field, indicating the potential for advancing oxide electronic devices.” are added.

3. Why did you choose this material? LaAlO₃/SrTiO₃, how about other similar materials?

Our response: LaAlO₃/SrTiO₃ has a strong correlation effect, and the two-dimensional electron gas can be controlled simply by an electric field, which shows the prospect of developing oxide electronic devices, as discussed in the introduction of our manuscript. Further, the construction of the device does not require complex micro-nano processing, which makes it possible to greatly reduce the cost [N. Reyren *et al.*, *Physical Review Letters* **108**, 186802 (2012)]. Furthermore, the high dielectric constant of STO makes it ideal for gating experiment. According to the parallel plate capacitor model, the depleted carrier concentration $n = Q/S = CV/S = \epsilon_r \epsilon_0 V/d = \epsilon_r \epsilon_0 E$. Where Q is the amount of charge in the gas ice layer, S is the electrode area, C is the capacitance, ϵ_r is the relative dielectric constant, ϵ_0 is the vacuum dielectric constant and E is the electric field intensity. The very high dielectric constant of STO thus allows us to tune the carriers over a very wide range.

In addition, the lower the carrier concentration, the better the device performance for low partial pressure according to our model. Carrier concentration can be well controlled in LAO/STO system by either growth condition (e.g., oxygen partial during growth [S. Thiel *et al.*, *Science* **313**, 1942 (2006)] or delta doping [T. Fix *et al.*, *Applied Physics Letters* **94**, 172101 (2009)]). With LFO delta doping, we indeed achieved very low carrier concentration. All these features make the LAO/STO system highly attractive for device applications, including gas sensors.

Inspired by the reviewer’s comments, we believe that this mechanism is also feasible

for other similar 2DEG system, such as some III-V semiconducting materials. Although we have not conducted studies on III-V materials, which are outside our research area, further research on III-V semiconductors within the scientific community is worth considering. It's worth noting that the relative dielectric constant of SrTiO₃ at low temperature (≤ 25 K) is three orders of magnitude larger than that of III-V materials like GaAs and GaN [O. G. Vendik *et al.*, *Journal of Superconductivity* **12**, 325 (1999) & M. J. Coak *et al.*, *Physical Review B* **100**, 214111 (2019)]. If the same carrier concentration is depleted, AlGaAs/GaAs and AlGaN/GaN systems would require an electric field (E) that is three orders of magnitude more intense than what is needed for LaAlO₃/SrTiO₃. In the other way, this situation may suggest that other 2DEG systems with low dielectric constant would need to achieve lower carrier concentrations to exhibit similar levels of performance.

4. In order to evaluate the structure of two-dimensional gas (C-2DEG) at a delta-doped LaAlO₃/SrTiO₃ interface with LaFeO₃, and more structural characterization should be considered, such as XPS, XRD, FTIR and so on.

Our response: The XRD structure characterization can be found in Fig. R4 (Fig. S2 in the SI). Both STEM and XRD results demonstrate that our film has a good epitaxial structure. In response to the reviewer's comment, we have also added Fe L-edge XAS data (Figure R5) to the SI (Fig. S3) in the revised manuscript, providing additional characterization data for our samples. Since the capped LAO layer blocks the release of excited electrons, the electron yield decreases, leading to a reduction of total electron yield signal and thus intensity in the underneath Fe XAS. The XAS results do not show an evident change of the valence state of Fe. In addition, there are some related researches already reporting detailed structure characterization of LFO/STO interface, such as G. J. Omar *et al.*, *Nano Letters* **20**, 2493 (2020) & P. Xu *et al.*, *Advanced Materials* **29**, 1604447 (2017).

Fig. R4 | Structure characterizations. **a**, X-ray diffraction of (001) and (002) peaks for the heterojunction film. **b**, X-ray diffraction ω -rocking curve of the as-grown heterojunction film (002) plane. Experimental (black dots) and model (red line) X-ray reflectivity curves (**c**) of the heterojunction film. **d**, X-ray scattering length density (SLD) profiles obtained from the X-ray reflectivity fits.

Fig. R5 | X-Ray absorption spectra (XAS) of Fe L-edge (Fe 2p). The XAS of the Fe L-edge of the heterojunction (red line, 10LaAlO₃/5LaFeO₃/SrTiO₃) and the single LaFeO₃ film (black line, 10LaFeO₃/SrTiO₃).

5. Although Figure 4 exhibited the possible mechanism of broad-spectrum C-2DEG gas sensor, the deeper mechanism still needs to be discussed again, and especially for the effect of temperatures.

Our response: The parallel-plate capacitor-like model that we proposed fits the existing data perfectly, suggesting a certain depth to the mechanism. In addition, we have added a discussion of the relative dielectric constant of the SrTiO₃ substrate in our manuscript, which closely aligns with the reported results. This result further supports our proposed model.

The corresponding description has been added on page 10 as follows. “In addition, the depleted carrier concentration can be expressed as $n = \epsilon_r \epsilon_0 V/d = \epsilon_r \epsilon_0 E$. Here, ϵ_r is the relative dielectric constant, ϵ_0 is the vacuum dielectric constant and E is the electric field intensity. Utilizing this model, the relative dielectric constant of our STO substrate is estimated to be 9600 at 2 K, which is close to actually ϵ_r of single crystal STO^{28,29}, further supporting our proposed model of parallel plate capacitors.”

Our next work uses different voltage (electric field intensity) control, and the results of switching devices with different resistance switching ratios also support this model. Some data are shown in the Fig. R6 below. The effect of temperature on the device has been discussed in Fig. R7 (Fig. 3i in the manuscript), revealing that the state of matter affects the chargeability.

However, this manuscript mainly demonstrates a novel broad-spectrum gas sensing mechanism and the performance of prototype devices. A more in-depth and accurate discussion of the mechanism would require additional in situ tests at low temperatures, such as secondary ion mass spectrometry and electrostatic measurements, etc. We need to seek more cooperation for follow-up work.

It is also worth to mention that the electronic state of the C-2DEG can be easily

controlled by an electric field to realize normally-on transistors with different resistance switching ratios. What is more, a low-temperature transistor for writing and erasing has been achieved through the coordinated control of the optical field and the electric field. Relevant data will be reported in the next paper, with a portion of the data shown in Fig. R6.

Figure R6 | Different voltages to achieve different non-volatile MIT with different resistance switch ratios.

Figure R7 | R - T curves of the device activated before (blue and green) and after (red and purple) the melting point of oxygen.

6. In order to evaluate the advantages of this work, similar studies should be compared and discussed.

Our response: To address the reviewer’s comment, we have added following sentence “Based on the underline physics for the gas sensing of our 2DEG system, we can anticipate similar outcomes for other 2DEG systems, Consequently, to achieve the same depleted carrier concentration, AlGaAs/GaAs and AlGaN/GaN systems necessitate an electric field (E) that is three orders of magnitude more intense than that required for the LaAlO₃/SrTiO₃ system.” on pages 10 & 11.

In addition, Part of the comparative work can be found in Table G1 (Table S2 in the SI).

Name	CO₂ measure limit	Author
C-2DEG Gas Sensor	0.025 torr (33 ppm at 1.0 atm)	Present work 2022
Calcium doped ZnO	50 ppm	Ghosh et al. 2019
p-Si/MoO₃	100 ppm	T. Thomas et al. 2021
Li₄Ti₅O₁₂	100 ppm	S. Joshi et al. 2016
CoAl₂O₄	100 ppm	C. Michel et al. 2010
ZnO	400 ppm	Y. Hunge et al. 2018
CeO₂	800 ppm	A. Aboud et al. 2017
SnO₂	2000 ppm	D. Wang et al. 2016
CN_x/p-Si	6 torr	N. Zouadi et al. 2015

Table G1 | A literature review on the CO₂ measure limit of gas sensor. The test limit of our C-2DEG gas sensor for CO₂ can be compared with the best MOS gas sensor that has been reported so far.

REVIEWER COMMENTS

Reviewer #1 (Remarks to the Author):

The authors addressed all the comments raised in my original review. The article is now clearer and more reachable with the additional data added by the authors. The article in its current form is suitable for publication in Nature Communications.

Reviewer #2 (Remarks to the Author):

The author tried to respond to reviewers' comments as best as possible. However, the discussion was not reflected in the paper.

Reviewer #3 (Remarks to the Author):

In the revised manuscript, the relevant questions have been answered in detail, and the quality of the article has been significantly improved. However, some doubts still need to be considered.

(1) Although LaAlO₃/SrTiO₃ has a strong correlation effect, the stability and self-supporting properties of 2D materials need to be evaluated, especially for the device stability and anti-interference ability.

(2) In addition, Table S2 listed the comparative work, while more performance parameters or conditions should be considered than just CO₂ measure limit.

(3) More explanations should be given for Figure R7.

REVIEWER COMMENTS

For Reviewer#2 (Remarks to the Author):

The author tried to respond to reviewers' comments as best as possible. However, the discussion was not reflected in the paper.

Our response: We greatly appreciate the reviewers' meticulous review. In response to their valuable feedback, we have incorporated the sensor's key highlights and unique characteristics into the revised manuscript and SI. Furthermore, we have introduced the challenges faced by current broad-spectrum gas sensors and elucidated our rationale for selecting LAO/STO in the introduction section.

(I) For the working principle of our devices, we have added more details to emphasize the device's advantages and clarify our statements.

(I - i) The tunable measurement range:

“Nonetheless, since the resistance at the low-temperature segment exceeds our electronic measurement range... .Therefore, the measurement range of such sensing devices can be rationally designed by controlling the carrier concentrations.” (on page 7 & 8 in the manuscript)

“The circular markers correspond to the sensor with a carrier concentration of $1.30 \times 10^{13}/\text{cm}^2$, while the square markers correspond to the sensor with a carrier concentration of $2.54 \times 10^{13} /\text{cm}^2$. The temperature difference associated with both the inflection points and the minimum value of the first-order differential of the two sensor resistances is less than 0.5 K within the error bars.” (on page 8 in the SI)

(I - ii) The lower activated electric field comparing with other 2-DEG systems:

“In addition, the depleted carrier concentration can be expressed as $n = \epsilon_r \epsilon_0 V/d = \epsilon_r \epsilon_0 E \dots$ AlGaAs/GaAs and AlGaN/GaN systems necessitate an electric field (E) that is three orders of magnitude more intense than that required for the LaAlO₃/SrTiO₃ system.” (on page 10 & 11 in the manuscript)

(I - iii) The sample structures in detail:

Due to space constraints, additional structural discussion and comparison, such as X-Ray absorption spectroscopy (XAS) of Fe L-edge, have been added to page 4 in the SI.

(II) For the practical applications, we also discussed the stability:

“In practical application scenarios, the stability of gas sensors is of paramount importance... .. This underscores the overall stability and cycle performance of our device.” (on page 11 & 12 in the manuscript)

We will provide a more comprehensive analysis of the detailed mechanism in our forthcoming work.

For Reviewer#3 (Remarks to the Author):

In the revised manuscript, the relevant questions have been answered in detail, and the quality of the article has been significantly improved. However, some doubts still need to be considered.

Our response: Thanks to the reviewers for his/her constructive comments to improve the quality of the article. Below we will address all raised comments point by point.

Comment 1: Although $\text{LaAlO}_3/\text{SrTiO}_3$ has a strong correlation effect, the stability and self-supporting properties of 2D materials need to be evaluated, especially for the device stability and anti-interference ability.

Our response: As mentioned in the introduction, C-2DEG is generated at the interface between two chemically stable insulators, LaAlO_3 and SrTiO_3 . This exceptional stability arises from its resistance to chemical reactions that involve electron gain or loss, making it highly chemically stable. Unlike traditional two-dimensional metallic films with complex packaging technology, C-2DEG is naturally buried at the interface between a substrate and an epitaxial thin film. As a result, the fabrication process is straightforward and does not necessitate complex self-supporting processes.

Furthermore, after hundreds of consecutive trials with the same device, the carrier concentration and mobility at the interface remained nearly constant, which guarantees the sustained stability of C-2DEG even after numerous back-gating processes.

Additionally, it is worth noting that previous research (S. Wu et al., *ACS Applied Materials & Interfaces* **6**, 8575 (2014).) has reported the stability of LAO/STO switch devices, demonstrating their durability with over 2,000 write/erase cycles. This underscores the overall stability and cycle performance of our device. Because such phenomenon is only decided by the state of the measured gas and DC measurements, the C-2DEG in our devices also could resist a lot of environmental disturbances, such as the electromagnetic interference.

On page 11 in the manuscript, the following sentence “In practical application scenarios,

the stability of gas sensors is of paramount importance... .. This underscores the overall stability and cycle performance of our device.” is added.

Comment 2: In addition, Table S2 listed the comparative work, while more performance parameters or conditions should be considered than just CO₂ measure limit.

Our response: In response to the reviewer's comment, we have listed other critical technique parameters for gas sensors of such as compatibility, working temperature, and response time in comparison table S2.

Name	CO₂ measure limit	Compatibility	Working temperature	Response time	Author
C-2DEG Gas Sensor	0.025 torr (33 ppm at 1.0 atm)	Broad-Spectrum	-273 °C – Room temperature	30 - 600 s (Depends on heating rate)	Present work 2023
Calcium doped ZnO	50 ppm	H ₂ & CO	350 °C	111 s	Ghosh et al. 2019[1]
p-Si/MoO ₃	100 ppm	Unknown	250 °C	8 s	T. Thomas et al. 2021[2]
Li ₄ Ti ₅ O ₁₂	100 ppm	Unknown	500 °C	5 s	S. Joshi et al. 2016[3]
CoAl ₂ O ₄	100 ppm	CO	450 °C	50 s	C. Michel et al. 2010[4]
ZnO	400 ppm	H ₂ & CO	350 °C	75s	Y. Hunge et al. 2018[5]
CeO ₂	800 ppm	H ₂ & CO	250 °C	Unknown	A. Aboud et al. 2017[6]
SnO ₂	2000 ppm	Unknown	240 °C	350 s	D. Wang et al. 2016[7]
CN _x /p-Si	3 torr	Unknown	Room Temperature	260 s	N. Zouadi et al. 2015[8]

Table S2. A literature review on the performance parameters and conditions of CO₂ gas sensor.

The following sentence:

“According to the parallel plate capacitor model, the device exhibits an instantaneous response when a substance undergoes sublimation or boiling. which is manifested as a change in the slope of the C-2DEG resistance at the interface. To precisely characterize

gas partial pressure, data should be acquired within a range of ± 10 K around the sublimation or boiling point. Consequently, the response time for measuring partial pressure depends on the heating rate (2 - 40 K/min). As a result, the response time can range anywhere from 30 to 600 s.”

has also been added on page 10 in the SI.

Comment 3: More explanations should be given for Figure R7.

Our response: A detailed discussion of Fig. R7 (Fig. 3i) is available on pages 8 and 9 in the manuscript.

For improved clarity, the sentence

“Therefore, we conclude that a temperature below the freezing point is sufficient to activate the C-2DEG gas sensor, and thus our C-2DEG gas sensor should be able to operate in a wide temperature range.”

on page 9 in the manuscript has been rephrased as follows:

“Therefore, the temperature mainly affects the activation of our C-2DEG gas sensor without affecting the working mode of the sensor, and just need be below the freezing point of the measured substances, permitting operation over a wider temperature range.”

REVIEWERS' COMMENTS

Reviewer #3 (Remarks to the Author):

In the revised manuscript and response letter, the relevant questions have been answered satisfactorily, and the quality of manuscript has been improved. Thus, I think it can be published without extra suggestion.